# How do e-commerce platforms and retailers implement discount pricing policies under consumers are strategic?

Hao Li[1,2], ZHe Chen[1]*

1 School of Economics and Management, Chongqing Jiaotong University, Chongqing, China, 2 Western China Transportation Economy-Society Development Studies Center, Chongqing, China

* chenzhe029@163.com

**Data Availability Statement:** All relevant data are within the manuscript and its Supporting Information files.

## Abstract

In the era of the rapid development of e-commerce, many retailers choose to launch promotional activities to become consumers' first choice for shopping. Since price discounts can greatly attract consumers, the e-commerce platforms have also begun to implement discount pricing. It is urgent for e-commerce platforms and retailers to formulate reasonable discount strategies to achieve sustainable business. In this paper, we construct a dynamic game model for implementing discount pricing on an e-commerce platform and two retailers, we study the market equilibrium between the two retailers and the e-commerce platform under various scenarios that considering consumers' strategic waiting behavior and competition between the two retailers, we further discuss the effectiveness of retailer discount pricing and the double discount pricing of the platform and retailers. We show that the optimal pricing decreases as the difference in product quality narrows under both pricing strategies. Low-quality retailers implementing a double discount pricing strategy are in relatively higher demand only when the difference in product quality is small. High-quality retailers implementing the retailer discount pricing strategy are in relatively higher demand only when the product quality difference is large. Double discount pricing is desirable for both e-commerce platforms and retailers and can be used to effectively achieve Pareto improvement in the market by increasing their expected profit. Our results emphasize the role of product quality and the value of the double discount pricing strategy. The double discount pricing strategy weakens the profit advantage that retailers and platforms gain from it as the rebate intensity and rebate redemption rates increase.

## 1. Introduction

In recent years, the online sales model organized by large e-commerce platforms with the participation of many online retailers has had a significant positive impact on the global economy. According to the latest data from eMarketer, the total global e-commerce transactions amounted to 4.938 trillion dollars in 2022. Online sales usually take the form of discount sales at specific times, such as "Black Friday" and "Cyber Monday" in the United States, "Boxing

**Funding:** This study was supported by the Chongqing Transportation Bureau's Science and Technology Project: Estimation of the Effect of Chongqing Transportation on Promoting Industrial Development, No. 2021-1. And the National Social Science Foundation project: Research on the Multi party Linkage Mechanism of the Governance of the Online Ride hailing Market under the Background of Co construction, Co governance, and Sharing, with the number of 19XGL016. The sponsor has no role in research design, analysis, publication decisions, or manuscript preparation.

**Competing interests:** The authors have declared that no competing interests exist.

Day" in Singapore, and "618" and "Double Eleven" in China. the purpose is to achieve customer penetration through shopping festivals. The emergence of these shopping seasons has greatly boosted the growth of product sales. For example, the huge consumer market in China has led to the rapid growth in online retail sales, which amounted to 2.56 trillion dollars accounting for 51.9% of the global sales. More than 300,000 brand retailers participated in Tmall "Double Eleven" in 2022, with sales exceeding 965.1 billion CNY on the day [1]. Online sales enable retailers to deal with customers more cost-effectively and efficiently than in traditional sales, while discount sales strategies under online sales provide e-commerce platforms and online retailers with a market means to enhance competitiveness, in order to improve market competitiveness under the network economy.

In the discount sales season, the product discounts enjoyed by consumers are often provided by e-commerce platforms or retailers in different forms on e-commerce platforms. In terms of online discount sales promotion, Amazon Japan launched promotional activities in advance of "Black Friday", including "50% off the whole site+ reward points", "free shipping" and "cash back". Similarly, Pinduoduo, one of the most famous e-commerce platforms in China, launched a "Billions Subsidies Promotion" to discount more than 23,000 products in 2019, and sold 400,000 units of the iPhone 11 on "Double Eleven" resulting in a 20-fold increase in sales of the iPhone series compared to the same period in the previous year. Alibaba started the 2022 Double Eleven early on 24 October and warmed up comprehensively through various platforms such as TikTok and Weibo. Meanwhile, retailers on the e-commerce platform issue coupons in advance in order to promote their promotion information more widely to expand their brand influence, and increase consumers' enthusiasm for participation, this includes "Full 300 minus 50", "Two for One", "Time-limited sec-killing", etc. Therefore, consumers can enjoy single discounts from retailers or double discounts from e-commerce platforms and retailers. The double discount is enough to send consumers on a shopping spree and drive a large wave of online spending.

In practical terms, various discount sales have a significant impact on consumers, leading to consumers' strategic waiting behavior. Currently, consumers are becoming more sophisticated and more strategic after years of discount sales. They tend to put their favorite products in their "shopping cart" before the sale and wait for discount to buy them to get a higher "thrill of the discount". The impact of such strategic consumer behavior on the retailer's profit performance can be highly detrimental, This strategic consumer behavior may have a very harmful impact on the profit performance of retailers, which may lead to many retailers sacrificing their participation in product profit for several months during discount sales to engage in price wars in exchange for increased sales, ultimately falling into a vicious cycle of "price reduction—price reduction". According to the statistics of "People's Daily Online of China", more than 90% of participating retailers in the massive discount sales on e-commerce platforms are just "loss leaders" [2]. Meanwhile, consumers who wait until the double discount sale period are faced with various promotional activities and complicated promotional rules on e-commerce platforms, such as "Presale", "Lazada bonus" and other prepurchase discount models as well as "Scratch cards", "Refund of deposit" and other postpurchase cashback models. They need to know various promotional rules in advance to participate in promotional activities to obtain coupons, so as to purchase the cheap and cheerful products they have long wanted. However, consumers may experience a "slip" phenomenon in the postrebate model, that is, consumers need to make efforts to obtain rebates, but may not redeem the rebate after purchasing the product. As a result, many promotions\discounts do not convert well. This has important implications for e-commerce platforms and retailers with double discount sales strategies.

In the context of the existing discount sales, exploring the rules of discount pricing and breaking the plight of retailers' bleak operation is the key to the successful operation of

enterprises and the sustainable development of discount sales seasons such as internet shopping festivals. How should e-commerce platforms and retailers implement their discount strategies in the face of consumers with strategic waiting behavior? It is therefore desirable to endogenous the strategic waiting behavior of consumers, capturing its effect on the pricing strategies of e-commerce platforms and retailers. To this end, we establish a two-stage pricing game model between two retailers and between two retailers and an e-commerce platform, and use this model to solve the equilibrium problem of the two-stage model. We attempt to answer the following questions:1) How do two retailers set optimal product prices to balance regular sales periods and discount sales periods? 2) In the case of rebates on the e-commerce platform, how can retailers change the market price and sales strategy considering that consumers enjoy product rebates on e-commerce platform? 3) What conditions would e-commerce platforms choose to provide rebates when retailers engage in two-stage discount pricing sales?

To address these questions, we propose the double discount pricing strategy based on traditional retailer discount pricing strategies, which differs from the traditional retailer discount pricing strategy in that we consider both e-commerce platforms and retailers providing discount to consumers. We compare the retailer discount pricing strategy to the double discount pricing strategy. These strategies are as follows: First, only the retailer implements discount pricing with a price reduction in the next period. Second, on the basis of the discount pricing implemented by the retailer, the e-commerce platform offers a rebate discount to the consumer, who can enjoy double discounts, one from the retailer and one from the e-commerce platform. Finally, we obtain the relevant factors that affect the profits of retailers and e-commerce platforms, and the applicable conditions of the different discount pricing strategies through equilibrium analysis.

This study contributes to the current literature in several aspects. First, we introduce consumers' strategic waiting behavior into pricing strategy research. We consider that some consumers may purchase products directly from manufacturers because they have a greater motivation to obtain products first. In contrast, some consumers may wait until the price of the product is reduced before purchasing the product. Second, we pay more attention to the actual competition situation of the platform, and competition problem between two retailers, we deeply explore the comparison between the discount pricing of the two retailers and the dual discount pricing of retailers and e-commerce platforms, we simplify the process of solving for the equilibrium profit of retailers and e-commerce platforms under different strategies, to clarify the optimal decision under different ranges. Third, since retailers' discounts and e-commerce platform rebate decisions have a certain impact on revenue, we analyze the effectiveness of the double discount pricing strategy compared with the traditional retailer discount pricing strategy, we also consider the consumer rebate redemption rate in the double discount pricing strategy. This provides some guidance for the pricing decisions of retailers and e-commerce platforms.

The remainder of this paper is organized as follows. Section 2 provides an overview of the relevant literature. Section 3 introduces the model and parameters. Section 4 introduces and analyzes the two-period dynamic game model of discount pricing between retailers and considers the case of double discount pricing when retailers engage in discount pricing while e-commerce platforms offer rebates, and analyzes the double discount pricing strategy equilibrium. Section 5 reports the numerical results and management insights of our model and analysis. Section 6 summarizes and highlights some future research directions. All proofs are included in the S1 Appendix.

## 2. Literature review

Our paper belongs to the stream of management science literature that studies the effects of strategic consumer behavior in the context of revenue management. We provide a review of a representative group. To keep the review of related research papers concise and effective, we split the presentation into several key aspects, including pricing decisions by retailers under consumer strategic waiting behavior, discount pricing strategies by retailers, and double discount pricing strategies by e-commerce platforms and retailers. We review relevant literature from the combination of these aspects and position our work relative to the literature.

One key aspect we address in this paper is consumers' strategic waiting behavior, a phenomenon that has been a topic of discussion in the behavioral economics and revenue management literature in recent years. Muth [3] and Coase et al. [4] demonstrate that it will contribute to a 20% increase in profits if the seller considers strategic consumer behavior in pricing decisions. Aviv [5], Otero [6], Li [7], Aflaki [8], and Zhang [9] focused on the pricing mechanism for retailers to deal with consumer strategic waiting behavior when there is only one price reduction point. These authors found that consumer strategic waiting behavior has a significant impact on retailers' pricing decisions, and retailers need to consider differences in strategic waiting willingness among consumers to effectively respond to market competition. Elmaghrab et al. [10] and Levin et al. [11] extended the number of price reductions to a finite number of times, studied the multi period pricing equilibrium problem for retailers under consumer strategic behavior, and developed a multi period dynamic pricing model to achieve market Pareto improvement. Su and Zhang [12] and Liu and Zhang [13] explored the dynamic pricing equilibrium problem of perishable goods considering consumers' price comparison behavior of intertemporal switching under a duopoly competition. Ozgun et al. [14] and Zhou et al. [15] studied pricing product decisions under strategic behavior and discussed the impact of this behavior on the performance of each node enterprise and supply chain performance.

The above studies fully illustrate the need for retailers to consider consumer strategic behavior in price strategies. Many retailers are beginning to attract consumers through price wars of discounted sales. Aviv and Pazgal [5] consider the problem of selling a fixed number of seasonal products to customers whose product valuations decline over time. Rhee and Thomadsen [16] found that if consumers offer sufficient discounts for future periods, companies at a competitive disadvantage will reward their customers, that is, if the difference in quality adjustment costs between the two companies is small, low-quality companies will reward their current customers, while if the cost difference is large, high-quality companies will reward their current customers. Jeong and Maruyama [17] found that a firm's optimal discounting strategy is to offer discounts to new customers in markets with more inertial consumers and higher switching costs, while firms offer discounts to past customers in a market where there are more variety-seeking consumers with large staying costs. Mantin and Veldman [18] consider consumers with strategic behaviors, and in the case of process improvement efforts, such supplier-initiated efforts ultimately reduce production costs and may translate into lower wholesale prices and lower consumer prices. Yang et al. [19] compared a two-advance-order-discount model with a one-advance-order-discount model and found the optimal decisions of upstream firms. In recent years, the discount sales strategy of the e-commerce platform in the online shopping festival further exacerbates the strategic behavior of consumers. It is worth researching and paying attention to how the coordination of the normal sales seasons and the discount sales seasons between the platform and retailers can be achieved when the discount sales strategy of e-commerce platforms includes considerations of consumer strategic behaviors. Lu and Moorthy [20] discussed the equilibrium pricing strategies under the discount sales strategies of coupons and quantity rebates provided by e-commerce platforms, and

compared the effectiveness of these two strategies in alleviating consumer strategic behavior, they found that consumers' risk preference is positively correlated with the effectiveness of two discount sales strategies. Mu et al. [21] constructed a Stackelberg game model of an e-commerce platform and a retailer to study the optimal pricing strategies of the retailer and the e-commerce platform under two scenarios: the e-commerce platform does not provide consumer rebates and provides consumer rebates; they show that when the hassle cost of consumer redemption rebates is high, the profit is higher when the e-commerce platform offers rebates compared to when it does not offer rebates. Liang et al. [22] divided product sales into an original price stage and a discount price stage and proposed a product pricing algorithm based on consumer strategic behavior to improve retailer profit by constructing a discount pricing model for a two-stage e-commerce platform. Mu et al. [23] analyzed the optimal rebate decisions of retailers and e-commerce platforms under the commission-driven rebate model and the marketing rebate model, indicating that both models contribute to mitigating strategic consumer behavior, helping retailers expand their market and increasing profit. Nie et al. [24] explored the optimal rebate pricing strategies of retailers under monopoly and competition by establishing four rebate models: upfront rebate, later rebate, always rebate, and never rebate; they showed that retailers would always choose the upfront rebate strategy under monopoly, while all four equilibrium rebate strategies may be effective in the competition situation.

By modeling and analyzing the two discount pricing strategies, we study the following two questions. First, we provide insight into the effectiveness of the two pricing strategies in mitigating strategic consumer behavior based on the double discount pricing strategies. Second, we identify the conditions under which the two discount pricing strategies can be most effectively used to mitigate strategic consumer behavior. Although Liu and Zhang [12], Lu and Moorthy [20], and other literature have studied the competition between two retailers offering differentiated products, but have not considered the impact of e-commerce platform discounts on retailers. Although Mu et al. [21] studies a double discount pricing strategy, but only consider one retailer and one e-commerce platform, without considering the competition among multiple retailers in the platform. Based on the research of Mu et al. [21], we not only consider the competition between two retailers, but also the competitive cooperation relationship between retailers and platforms. We analyze the effects of factors such as retailer competition and double rebates on market equilibrium pricing, e-commerce platforms, and retailers' expected profits under strategic consumer behavior. This approach can be used to better simulate market practices during the discounted sales season., to some extent, it further developed research on discount pricing strategies. In order to highlight the difference between retail discount pricing strategies and double discount pricing strategies, we reflect the rebate factor and the rebate exchange rate of consumers in the consumer utility function. To the best of our knowledge, few studies have simultaneously analyzed the impact of different discount pricing strategies on alleviating consumer strategic waiting behavior. Therefore, we consider retailers' single discount pricing. Then, we attempt to determine the effectiveness of these two discount pricing strategies by depicting consumers enjoying double discounts through e-commerce platform rebates.

## 3. The model

Consider a market with two competing retailers, retailer H and retailer L, and an e-commerce platform E. These two retailers sell products to consumers on e-commerce platforms, which charge commissions from the retailer's sales. The two retailers offer products that are perishable with differences in performance or service. The different products are characterized by a quality index $q_i$(i = H,L), and we assume $q_H > q_L$ throughout; therefore, product H has higher

quality. Without loss of generality, we normalize $(q_H, q_L)$ to $(1, \beta)$ with $\beta \in (0,1)$. Additionally, similar to Liu and Zhang [13] and Zhao [25], $\beta$ can be expressed as the consumer's evaluation of the differentiated product or product quality difference.

The selling season for the products is divided into two consecutive periods with the regular sales period (defined as Period 1) and the discount sales period (defined as Period 2). The products can be sold at different prices in different time periods. All customers arrive at the beginning of the sales season prior to Period 1. Retailers offer discounts and promotions on their products during the second period of price reduction. The total number of customers is normalized to 1, and customers purchase at most one during the entire selling season. Customers are intertemporal utility maximizers and have heterogeneous valuations of quality, which is denoted by v, we assume that v follows a uniform distribution on [0,1], which is common knowledge for the retailers and consumers. According to online shopping festival sales practices, there are two ways to provide discounts. One way is for retailers offer discounts to consumers, such as "get an extra 20% OFF" on the North Face outdoor brand and the Barbour windbreaker brand is offering a "up to 60% OFF" on Black Friday. The other is through e-commerce platforms providing consumers with various preferential rebate strategies in the practice of discount sales in addition to retailers; an example of this is the Zalora platform in Singapore, which offers "Rebate activities", which means that consumers can still obtain rebates after confirming the receipt of products despite having received a discount. During "Double Eleven" in China, Taobao offered a cross-store discount, the Tmall platform for the total amount of shopping launched the "Full 300 minus 30" allowance, and "6.18" on the JD platform set the Koi card to divide millions of red packets, etc. Therefore, we consider these two sales practices separately, which are defined as retail discount pricing strategies, where only high-quality retailers provide discounts to consumers, and the double discount pricing strategy of e-commerce platforms and retailers, where retailers provide discounts to consumers while e-commerce platforms provide rebates to consumers. We explore the effectiveness of these two discount pricing strategies in different product sales environments. The decisions of the e-commerce platform and retailers under a realistic discount sales season are shown in S1 Fig.

The rate of commissions paid by the retailer to the e-commerce platforms is $\delta$, which is mainly dependent on industry practices (Heese [26]); for example, the Shopee cross-border e-commerce platform sets a commission rate of 2%. The share that retailers receive from sales is $1-\delta$. And we denote the utility discount factor for strategic waiting by the parameter $\gamma$, $\gamma \in [0,1]$, a higher $\gamma$ means that larger values indicate greater intertemporal purchase utility and a higher willingness of consumers to wait until the discount sale period to purchase. Consumers have strategic behaviors and compare the utility of regular and discounted sales to make purchase choices. We need to consider the intertemporal choice behavior of consumers in a dynamic game framework, in which each customer decides on which product to purchase and when to purchase to maximize individual surplus. We can interpret $\gamma$ as the level of strategic behavior (Liu and Zhang [13]).

We assume that $\lambda$ is the rebate redemption rate. In reality, consumers who buy products due to rebate promotions may fail to get the rebates because they forget or miss the rebate period. Nowadays, the promotion methods of the platform are more diverse. For example, Tmall launched the pre-payment deposit booking product during the Double Eleven shopping festival. After paying the balance, consumers can directly apply for a refund of the deposit on the shopping details page to complete the rebate or "get coupon rebate for good reviews", offers on Taobao; consumers who buy the products do not get the rebate because they forget to apply for no deposit or fill out positive reviews. According to the Japan Times survey, only 40% of consumers are successful in points rebate activities. So, not all consumers can receive rebates provided by e-commerce platforms, thus we assume a rebate redemption rate $\lambda$.

Consumers often need to spend time and search costs to obtain rebates after purchasing products. We assume that the effort costs paid by consumers to obtain rebates at the two retailers are simplified to $c_1$ and $c_2$, respectively. The retailers aim to maximize their total expected profit in two periods, and they determined the price of product as $p_t^i(t = 1, 2)$. The price $p_2^i$ of the product purchased by consumers during the discount period is lower than the price $p_1^i$ of the product purchased during the normal period, which means that consumers can enjoy the discount of $1 - \frac{p_2^i}{p_1^i}$. For example, the Chinese shopping festival "Double Eleven" refers to the price reduction sales starting on November 11th. Before November 11th, the price of the Arthur ASICS brand sports shoe Flux 4 was 559 CNY, and the price of the product was reduced to 332 CNY from 00:00 on November 11th, This means that consumers enjoy a 41% discount. Depending on the relationship between $p_1^i$ and $p_2^i$ and the values of β and λ, retailers face a demand of $q_t^i$. The utility of the product purchased by the consumer in Period t is $U(v, p_t^i) = v - p_t^i$, and the value $v_T \in [0, 1]$ is the valuation of a marginal customer who is indifferent toward purchasing in either Period 1 or Period 2. The total profit of the two retailers over the sales period is $\pi^i = \sum_{t=1}^2 \pi_t^i = \sum_{t=1}^2 p_t^i q_t^i$, where I = H,L. We construct the dynamic pricing models for the two periods under retailer discount pricing and double discount pricing strategies for e-commerce platforms and retailers respectively, and the solution concept we adopt is the Markov perfect equilibrium (MPE), which is a profile of Markov strategies that is subgame perfect for each player. Let us define the strategy that only retailers conduct discount pricing in Period 2 as strategy E and the double discount pricing strategy that retailers provide discounts while e-commerce platforms provide rebates in Period 2 as strategy F.

## 4. Model analysis and results

### Analysis of retailer discount pricing strategy

In strategy E, only the retailer offers the discount, so retailer L may or may not incur demand in Period 2. Let $U_2^L = \beta v_2^L - p_2^L = 0$, $v_2^L$ satisfies $v_2^L = \frac{p_2^L}{\beta}$, $\cdot v_2^L$ indicates that there is no difference in the utility between consumers who do not purchase products or purchase products from retailer L. When retailer L incurs positive demand, the marginal valuation is determined by comparing the surpluses of purchasing L in Period 1 and purchasing H in Period 2; that is, when $U_2^L > U_2^H$ and $U_2^L \geq 0$, a customer with valuation $v \leq v_T$ will purchase from retailer L. Similarly, a customer with valuation $v \leq v_T$ purchases from retailer H when $U_2^H > U_2^L$ and $U_2^H \geq 0$, Let $v_T^*$ be the valuation that the utility of the product purchased by the consumer in the normal sales period and the discount sales period is equal. The indifference point of demand of two retailers in Period 2 satisfies $U_2^H = U_2^L$, that is $v_2^H = \frac{p_2^H - p_2^L}{1-\beta}$. The payoff function of retailer H and retailer L is:

$$\pi_2^H = p_2^H \left( v_T - \frac{p_2^H - p_2^L}{1 - \beta} \right) \tag{1}$$

$$\pi_2^L = p_2^L \left( \frac{p_2^H - p_2^L}{1 - \beta} - \frac{p_2^L}{\beta} \right) \tag{2}$$

The following proposition characterizes the equilibrium in Period 2.

**Proposition 1:** Under the retailer discount pricing strategy, the equilibrium prices in Period 2 are $p_2^{H*} = \frac{2(1-\beta)v_T}{4-\beta}$ and $p_2^{L*} = \frac{\beta(1-\beta)v_T}{4-\beta}$, respectively. The equilibrium expected profits of the two

retailers and the e-commerce platform are given by $\pi_2^{H*} = \frac{4(1-\beta)}{(4-\beta)^2}\left(v_T\right)^2$, $\pi_2^{L*} = \frac{(1-\beta)\beta}{(4-\beta)^2}\left(v_T\right)^2$ and $\pi_2^{E*} = \delta\frac{4-3\beta-\beta^2}{(4-\beta)^2}\left(v_T\right)^2$, respectively.

The proof of Proposition 1 is provided in the S1 Appendix. Recall that β is the relative quality level of the two products, and δ represents the rate of commissions. Proposition 1 demonstrates that the impact of β and δ on prices can lead to changes in equilibrium outcome.

Next, we analyze consumers who purchase a product from retailer H in Period 1, satisfying $v - p_1^H \geq \beta v - p_1^L$, $v - p_1^H \geq \gamma\left(v - p_2^{H*}\right)$, $v - p_1^H \geq \gamma\left(\beta v - p_2^{L*}\right)$. Similarly, consumers purchase a product from a retailer L, satisfying $\beta v - p_1^L \geq v - p_1^H$, $\beta v - p_1^L \geq \gamma\left(\beta v - p_2^{L*}\right)$, $\beta v - p_1^L \geq \gamma\left(v - p_2^{H*}\right)$. Therefore, the indifference point between consumer purchases at retailer H and retailer L in Period 1 is $v_1^H = \frac{p_1^H - p_1^L}{1-\beta}$. Let $v_T^*$ be the valuation of a marginal customer who is indifferent toward purchasing in either Period 1 or Period 2, satisfying $\beta v_T^* - p_1^L \geq \gamma\left(v_T^* - p_2^{H*}\right)$ i.e., $v_T^* = \frac{p_1^L(4-\beta)}{(4-\beta-\gamma)\beta-2\gamma}$. The payoff functions of retailers H and L in Period 1 are:

$$\pi^H = p_2^H\left(v_T^* - v_2^H\right) + p_1^H\left(1 - v_1^H\right) \tag{3}$$

$$\pi^L = p_2^L\left(v_2^H - v_2^L\right) + p_1^L\left(v_1^H - v_T^*\right) \tag{4}$$

**Proposition 2:** Under the retailer discount pricing strategy, the equilibrium prices for the two retailers in Period 1 are given by:

$$p_1^{H*} = \frac{2(1-\beta)\left((6-\gamma^2+\gamma)\beta^2 - \gamma\beta^3 - (15+4\gamma^2-10\gamma)\beta - 4\gamma^2 + 8\gamma\right)}{\beta^4 - (2\gamma+8)\beta^3 + (40-3\gamma^2)\beta^2 - (60+12\gamma^2-24\gamma)\beta - 12\gamma^2 + 32\gamma} \tag{5}$$

$$p_1^{L*} = \frac{(1-\beta)\left(\beta^2 - (4-\gamma)\beta + 2\gamma\right)^2}{\beta^4 - (2\gamma+8)\beta^3 + (40-3\gamma^2)\beta^2 - (60+12\gamma^2-24\gamma)\beta - 12\gamma^2 + 32\gamma} \tag{6}$$

with the corresponding equilibrium profits of: $\pi_1^{H*} = p_1^{H*}\left(1 - \frac{p_1^{H*}-p_1^{L*}}{1-\beta}\right) + \pi_2^{H*}\left(v_T^*\right)$, $\pi_1^{L*}$, $\pi_1^{L*} = p_1^{L*}\left(\frac{p_1^{H*}-p_1^{L*}}{1-\beta} - \frac{p_1^{L*}(4-\beta)}{(4-\beta-\gamma)\beta-2\gamma}\right) + \pi_2^{L*}\left(v_T^*\right)$, $\pi_1^{E*} = \delta\left(\pi_1^{H*} + \pi_1^{L*}\right)$.

The proof of Proposition 2 is provided in the S1 Appendix. Proposition 2 demonstrates the critical role that β and γ play in determining the equilibrium outcome. The detailed relationship between them can be found in numerical simulation.

## Analysis of the double discount pricing strategy for retailers and e-commerce platforms

Consumers can enjoy double discounts at the same time. On the one hand, retailers themselves offer discounts on products, and on the other hand, the platform provides category coupons, which can be shared. For example, if the price of a T-shirt on Taobao is 100 CNY, the shop is holding a promotion of subtracting 20 CNY from 100 CNY, and the platform presents a coupon of subtracting 10 CNY from 100 CNY for the product, then the customer can enjoy both the retailer's discount and the platform's coupon. Therefore, retailers set preferential product prices and e-commerce platforms offer rebates to consumers under the double discount pricing strategy of e-commerce platforms and retailers. To simplify presentation, we define the decision variable under the double discount pricing strategy of the platform and the retailer is superscript "~". And the rebate rate of the e-commerce platform as "f", this means that consumers can enjoy a discount of (1-f) on the basis of the original pricing in Period 2, amounting

to the actual price paid f$\tilde{p}_2^i$ under both platform and retailer discounts. We directly reflect the rebate paid by the e-commerce platform on the price paid by consumers, so as to more intuitively show the changes in consumer spending before and after the platform rebate. These notations will be used throughout the paper.

In strategy F, the consumer utility considering the consumer rebate redemption rate is formulated as $\tilde{U} = \lambda(\tilde{v} - f\tilde{p}_t^i - c_t) + (1-\lambda)(\tilde{v} - \tilde{p}_i^i)$. Consumers purchase at retailer L in Period 2 if $\tilde{U}_2^L \geq \tilde{U}_2^H$, that is, $\lambda(\beta\tilde{v} - fp_2^L - c_2) + (1-\lambda)(\beta\tilde{v} - \tilde{p}_2^L) \geq \lambda(\tilde{v} - f\tilde{p}_2^H - c_1) + (1-\lambda)(\tilde{v} - \tilde{p}_2^H)$, $\tilde{p}_2^H \geq \frac{\tilde{v}_T - \beta\tilde{v}_T + \tilde{p}_2^L - \lambda\tilde{p}_2^L + f\lambda\tilde{p}_2^L - c_1\lambda + c_2\lambda}{1 - \lambda + f\lambda}$, $\tilde{p}_2^H \geq \frac{(1-\beta)\tilde{v}_T - (c_1-c_2)\lambda}{1-(1-f)\lambda} + \tilde{p}_2^L$. Similarly, consumers purchase at retailer H in Period 2 if $\tilde{U}_2^H \geq \tilde{U}_2^L$, i.e., $\tilde{p}_2^L \geq \tilde{p}_2^H - \frac{(1-\beta)\tilde{v}_T - (c_1-c_2)\lambda}{1-(1-f)\lambda}$. Let $\tilde{v}_T$ be the valuation of a marginal customer who is indifferent toward purchasing in either Period 1 or Period 2. Let $\tilde{U}_2^H = \tilde{U}_2^L$, the point of indifference between consumers purchasing products from retailer H and retailer L is $v_2^H = \frac{\tilde{p}_2^H - \tilde{p}_2^L - \lambda((1-f)(\tilde{p}_2^H - \tilde{p}_2^L) - c_1 + c_2)}{1-\beta}$, and let $\tilde{U}_2^L = 0$, the point of indifference between consumers buying products and not buying products at retailer L is $v_2^L = \frac{(1-\lambda+f\lambda)\tilde{p}_2^L + c_2\lambda}{\beta}$. The profits of retailers H, L and the platform in Period 2 are:

$$\tilde{\pi}_2^H = \tilde{p}_2^H\left(\tilde{v}_T - \tilde{p}_2^H - \tilde{p}_2^L - \lambda\left(\frac{(1-f)(\tilde{p}_2^H - \tilde{p}_2^L) - c_1 + c_2}{1-\beta}\right)\right) \tag{7}$$

$$\tilde{\pi}_2^L = \tilde{p}_2^L\left(\frac{\tilde{p}_2^H - \tilde{p}_2^L - \lambda((1-f)(\tilde{p}_2^H - \tilde{p}_2^L) - c_1 + c_2)}{1-\beta}\right) - \frac{(1-\lambda+f\lambda)\tilde{p}_2^L + c_2\lambda}{\beta}) \tag{8}$$

$$\tilde{\pi}_2^E = \delta\left(\tilde{\pi}_2^H + \tilde{\pi}_2^L\right)$$

**Proposition 3:** Under the double discount pricing strategy for e-commerce platforms and retailers, the equilibrium prices in Period 2 are $\tilde{p}_2^{H*} = \frac{(c_2 - (2-\beta)c_1) + 2(1-\beta)\tilde{v}_T}{(1-(1-f)\lambda)(4-\beta)}$ and $p_2^L = \frac{((c_1+c_2)\beta - 2c_2)\lambda + \beta(1-\beta)\tilde{v}_T}{(1-(1-f)\lambda)(4-\beta)}$, respectively. The equilibrium expected profits of the two retailers and the e-commerce platform are $\tilde{\pi}_2^{H*} = \frac{4\left(\left((1-\frac{1}{2}\beta)c_1 - \frac{1}{2}c_2\right)\lambda - \tilde{v}_T(1-\beta)\right)^2}{(1-(1-f)\lambda)(4-\beta)^2(1-\beta)}$, $\tilde{\pi}_2^{L*} = \frac{\left(\beta^2\tilde{v}_T - \beta((c_1+c_2)\lambda + \tilde{v}_T) + 2c_2\lambda\right)^2}{\beta(1-(1-f)\lambda)(4-\beta)^2(1-\beta)}$ and $\tilde{\pi}_2^{E*} = \delta\left(\tilde{\pi}_2^{H*} + \tilde{\pi}_2^{L*}\right)$, respectively.

Let $\tilde{U}_1^H = \tilde{U}_1^L$, the indifference point of in utility between the two retailers in Period 1 is $\tilde{v}_1^H = \frac{\left((c_2-c_1)\lambda - A_H\tilde{p}_1^L\right)\beta^2 + \left(3A_H\tilde{p}_1^L + B_H\right)\beta + 2\tilde{p}_1^L - \left((2-2f)\tilde{p}_1^L + 2\gamma c_1 - 2c_2\right)\lambda}{(1-\beta)\left((4-\gamma)\beta - \beta^2 - 2\gamma\right)}$. To simplify the presentation, we define $A_H = (1-f)\lambda - 1$, $B_H = ((c_1+c_2)\gamma + 2c_1 - 4c_2)\lambda$. Similarly, $\tilde{v}_2^H$ is the indifference utility point of the two retailers in Period 2, and $\tilde{v}_3^L$ is the indifference point of a consumer purchasing at retailer L or not in Period 2. The indifference point valuation between two periods is $\tilde{v}_T^*$, that is, $\tilde{v}_T^* = \frac{\left((c_2-(1-f)\tilde{p}_1^L)\beta + (4-4f)\tilde{p}_1^L + (2c_1+c_2)\gamma - 4c_2\right)\lambda - (4-\beta)\tilde{p}_1^L}{\beta^2 - (4-\gamma)\beta + 2\gamma}$. Consumers purchase at retailer H in Period 1 if $\lambda(\tilde{v} - f\tilde{p}_1^H - c_1) + (1-\lambda)(\tilde{v} - \tilde{p}_1^H) \geq \lambda(\beta\tilde{v} - f\tilde{p}_1^L - c_2) + (1-\lambda)(\beta\tilde{v} - \tilde{p}_1^L)$, $\lambda(\tilde{v} - f\tilde{p}_1^H - c_1) + (1-\lambda)(\tilde{v} - \tilde{p}_1^H) \geq \gamma(\lambda(\tilde{v} - f\tilde{p}_1^H - c_1) + (1-\lambda)(\tilde{v} - \tilde{p}_2^{H*}))$, $\lambda(\tilde{v} - f\tilde{p}_1^H - c_1 + (1-\lambda)(\tilde{v} - \tilde{p}_1^H) \geq \gamma(\lambda(\beta\tilde{v} - f\tilde{p}_2^{L*} - c_2) + (1-\lambda)(\beta\tilde{v} - \tilde{p}_2^{L*}))$. Consumers purchase at retailer L in Period 1 if $\lambda(\beta\tilde{v} - f\tilde{p}_1^L - c_2) + (1-\lambda)(\beta\tilde{v} - \tilde{p}_1^L) \geq \lambda(\tilde{v} - f\tilde{p}_1^H - c_1) + (1-\lambda)(\tilde{v} - \tilde{p}_1^H)$, $\lambda(\beta\tilde{v} - t\tilde{p}_1 L - c_2) + (1-\lambda)(\beta\tilde{v} - \tilde{p}_1^L) \geq \gamma(\lambda(\beta\tilde{v} - f\tilde{p}_2^{L*} - c_2) + (1-\lambda)(\beta\tilde{v} - \tilde{p}_2^{L*}))$, $\lambda(\beta\tilde{v} - f\tilde{p}_1^L - c_2) + (1-\lambda)(\beta\tilde{v} - \tilde{p}_1^L) \geq \gamma(\lambda(\tilde{v} - f\tilde{p}_2^{H*} - c_1) + (1-\lambda)(\tilde{v} - \tilde{p}_2^{H**}))$.

Let $\tilde{v}_1^H$ be the valuation of customers who are indifferent to purchasing at retailers H and L in Period 1, satisfying
$\lambda\left(\tilde{v}_1^H - f\tilde{p}_1^H - c_1\right) + (1-\lambda)\left(\tilde{v}_1^H - \tilde{p}_1^H\right) = \lambda\left(\beta\tilde{v}_1^H - f\tilde{p}_1^L - c_2\right) + (1-\lambda)\left(\beta\tilde{v}_1^H - \tilde{p}_1^L\right)$. The payoff functions of H and L are:

$$\tilde{\pi}^H = \tilde{p}_1^H\left(1 - \tilde{v}_1^H\right) + \tilde{\pi}_2^H \tilde{v}_T^* \tag{9}$$

$$\tilde{\pi}^L = \tilde{p}_1^L\left(\tilde{v}_1^H - \tilde{v}_T^*\right) + \tilde{\pi}_2^L \tilde{v}_T^* \tag{10}$$

The proof of Proposition 3 is provided in the S1 Appendix. The following proposition characterizes the equilibrium in Period 1.

**Proposition 4:** Under the double discount pricing strategy for e-commerce platforms and retailers, the equilibrium prices in Period 1 are:

$\tilde{p}_1^{H*} = \dfrac{(c_1\lambda - 2\gamma)\beta^4 + D_H\beta^3 + D_L\beta^2 + E_H\beta - 8\gamma\left(\left(\left(c_1 - \frac{1}{4}c_2\right)\lambda - 1\right)\gamma + 2 + \left(\frac{3}{4}c_2 - 2c_1\right)\lambda\right)}{\left(-\beta^4 + 2\beta^3\gamma + 3\beta^2\gamma^2 + 8\beta^3 + 12\beta\gamma^2 - 40\beta^2 - 24\beta\gamma + 12\gamma^2 + 60\beta - 32\gamma\right)(1 - \gamma + f\gamma)}$ and

$\tilde{p}_1^{L*} = \dfrac{-\beta^5 + ((c_1 + c_1)\lambda - 2\gamma + 9)\beta^4 + E_L\beta^3 + F_H\beta^2 + F_L\beta - 4\gamma(((c_1 + 2c_2)\gamma - 5c_2)\lambda - \gamma)}{3(1 - (1-f)\lambda)\left(-\frac{1}{3}\beta^4 + \left(\frac{2}{3}\gamma + \frac{8}{3}\right)\beta^3 + \left(\gamma^2 - \frac{40}{3}\right)\beta^2 + \left(4\gamma^2 - 8\gamma + 20\right)\beta + 4\gamma^2 - \frac{32}{3}\gamma\right)}$, respectively.

Where $D_H = -2\gamma^2 + (4 + (2c_1 + 2c_2)\lambda)\gamma + 12 - (10c_1 + c_2)\lambda$,
$D_L = ((c_1 + 2c_2)\lambda - 6)\gamma^2 + (18 - (10c_1 + 6c_2)\lambda)\gamma - 42 + (30c_1 - 2c_2)\lambda$,
$E_H = -2\lambda\left(c_1 - \frac{5}{2}c_2\right)\gamma^2 + ((10c_1 - 8c_2)\lambda - 4)\gamma + 30 - (30c_1 - 12c_2)\lambda$,
$E_L = ((6c_1 + 2c_2)\gamma - 12c_1 - 10c_2)\lambda - \gamma^2 + 6\gamma - 24$,
$F_H = ((5c_1 + c_2)\gamma^2 - (20c_1 + 12c_2)\gamma + 20c_1 + 36c_2)\lambda - 3\gamma^2 + 12\gamma + 16$, and
$F_L = ((8c_1 - 2c_2)\gamma^2 - (4c_1 - 8c_2)\gamma - 36c_2)\lambda - 16\gamma$.

The proof of Proposition 4 is provided in the S1 Appendix. It can be seen that the pricing and expected profit of the retailer under the double discount sales strategy are significantly different from the e-commerce platform's strategy of not providing rebates. The analysis of the equilibrium results is potentially very complex because each parameter can take any value in [0,1], and complex parameter results are difficult to compare. Therefore, the following section numerically simulates the effects of product variation factor β, rebate rate f, and rebate redemption rate λ on the equilibrium price and profit, we analyzes the property of the equilibrium decision of the retailer and e-commerce platform under the two pricing strategies. These results are established in the Numerical Study and Managerial Insights section.

## 5. Numerical study and managerial insights

In this section, we conduct exhaustive numerical experiments and discuss the managerial implications of our models and results. Our experimental design centers around four main contributing factors in the model, the vertical product differentiation β, the rebate rate f, the redemption rate λ, the level of strategic behavior γ. We consider different values of β while fixing $\gamma = 0.4$, $f = 0.7$, $\lambda = 0.4$, $c_1 = 0.05$, and $c_2 = 0.03$, and we consider different values of f, λ and γ while fixing $\beta = 0.5$. According to the industry practice of online sales platforms, the commission percentage factor is $\delta = 0.2$, and similar values are used in the paper by Zhou [15]. After comparing multiple sets of values, we select the value with the most significant trend for display. Other numerical values may change the specific value of corresponding price, demand and income, but have no impact on the changes in factor relationships.

### Optimal pricing strategy analysis

The comparative analysis of the influence of product quality difference coefficient β, rebate rate f, rebate redemption rate λ and the level of strategic behavior γ on the equilibrium on the two-stage optimal pricing in the sales period under two different scenarios, namely, retailer

discount pricing strategy and e-commerce platform double discount pricing strategy, is shown in S2 and S3 Figs.

From the above figure, we can see that the smaller the difference is between the product quality offered by the two retailers (with the increase in β), the equilibrium price of retailer H in the two periods and the equilibrium price of retailer L in Period 2 decrease. The optimal pricing of retailer L in Period 1 increases and then decreases. Due to our assumption that the production and sales costs of retailers are 0, the prices in the above figure gradually tend towards cost. In essence, the reason is that retailers face homogeneous competition, and have to carry out "price wars" to attract consumers to buy products due to the increasing degree of product quality differences. This means that retailers should not narrow product quality differences to capture the market when using discount sales, but maintain a certain degree of product differentiation to focus on market segments, which is conducive to alleviate price competition among retailers. Then the optimal pricing for the retailer in both periods under the double discount pricing strategy is higher than the retailer discount pricing, which indicates that the essence of the platform rebate under the double discount pricing strategy is to give consumers an "illusion" of discounts. The internet shopping festival business "increase before decrease" routine has been frequently exposed in recent years. For example, the price of some brands of mobile phones had fallen to the low 2000 CNY on the eve of the shopping festival, but the price rose instead of falling after the Double 11, over 2000 CNY by several hundred. With such inflated prices and disorderly discounts, consumers are also less willing to buy, which affects retailers' sales.

The prices of the two retailers fall as the rebate rate increases and rise as the rebate redemption rate increases. An increase in the rebate rate f means that the discount (1-f) enjoyed by consumers on the platform will be reduced. In order to make up for the weakened discount enjoyed by consumers on the platform, retailers will choose higher discount and price reduction to attract consumers, while when the rebate redemption rate increases and consumers receive higher discounts from the platform, the retailer will take the opportunity to increase prices to gain higher revenue. The two retailers increase their prices with the increase of γ in Period 2 because as consumers become more willing to wait for the lower price of the second period, the retailers who cannot afford too many discounts will choose to increase their prices to ensure return.

### The impact of product quality differences on demand

The comparative analysis of the influence of the product quality difference coefficient β on the two-stage demand and total demand in the sales period under two different scenarios, namely, strategy E and strategy F, is shown in S4–S6 Figs.

We observe that for a fixed f value, both retailers' demand in Period 1 is increasing in β and the demand in Period 2 is decreasing in β. When the degree of product quality differences is small, the demand of both retailers under the strategy E is higher than that under the strategy F; when the degree of product quality differences is moderate, retailer H has higher demand for implementing the strategy F, and retailer L has higher demand for implementing strategy E; when the degree of product differences is small, the strategy F can drive higher demand from both retailers. For a larger β, the two retailers can obtain similar performance incentives under the double discount strategy. When the degree of product differences is moderate, consumers face the same discount activities and think that it is more cost-effective to buy high-quality products at that time, because they feel that they are "taking advantage" when comparing the product with the previous price, so high-quality retailer H can achieve consumer penetration under the platform subsidy, while low-quality retailers L can only carry out break-even

promotions and cannot obtain considerable sales under the double discount strategy. For a smaller β, low-quality retailers are more reluctant to participate in double discount pricing in the discount sales season, while the two retailers have greater demand under the retailer discount pricing strategy and have differentiated market segment advantages.

## The impact of product quality differences on profits

The comparative analysis of the product quality difference coefficient β on the profit of two retailers and e-commerce platforms under two strategies, is shown in S7–S10 Figs.

These figures illustrate the equilibrium outcomes between retailers and e-commerce platform. For both of them, the expected profit of the double discount pricing strategy outperforms that of the retailer discount pricing strategy for different degrees of product quality differences. We show that under the double discount pricing strategy, retailers will increase the price before the discount sales season, but with the e-commerce platform rebate subsidy and retailer discount pricing, this change can erroneously give the consumer a sense of increased utility, thereby increasing profits for both the platform and retailers. We believe that retailers and e-commerce platforms should adopt a double discount pricing strategy. However, as the degree of difference in product quality increases, the degree of profit growth between retailers and platforms decreases, which shows that homogeneous competition weakens the profit advantage obtained from the rebates of e-commerce platforms.

Moreover, we can also see from the above figure that the smaller the degree of product quality differences is, the more intense the competition between the two retailers will be. Offering two products with little difference and the same platform rebates will lead to a convergence of pricing between the two retailers, leading to the accumulation of homogeneous competition for retailers and lower profit. Therefore, retailers should maintain certain product quality differentiation. Because we are in an era of product excess, products tend to be homogeneous, but consumer needs have become more diverse and personalized, consumers only buy products they like and truly want. Therefore, maintaining a certain degree of product quality differentiation plays a key role in the competition to attract and retain consumers.

To further verify the effectiveness of the double discount pricing strategy, we denote the profit margin between the two pricing strategies by $\Delta \pi^i = \tilde{\pi}^i - \pi^i (i = H, L, E)$. We numerically verified the impact of the rebate rate f and rebate redemption rate λ on the profit of retailers and e-commerce platforms under the two pricing strategies. It can be obtained by the magnitude of $\Delta \pi^i$; see S11 Fig for illustration.

Our numerical studies indicate that $\Delta \pi^H, \Delta \pi^L$, and $\Delta \pi^E$ are always greater than zero for different degrees of rebate rates f and rebate redemption rates λ, which indicates that the double discount pricing strategy is always effective for the retailer as well as for the e-commerce platform. We found that the profit margin between the retailers and the platform decreases with the increase in the rebate rate, and the impact of rebate redemption rates on the profit is the same. The greater rebate rate f means a smaller discount (1-f) for consumers on the platform, so the positive impact of rebates on demand growth diminishes and the advantage of a double discount pricing strategy weakens accordingly. Rising redemption rates indicate that the e-commerce platform needs to pay a larger rebate value and that the role of the rebate efficiency advantage is gradually decreasing. The increase in profits from an increase in market share cannot compensate for the loss caused by paying a higher rebate value. This also suggests that the profit advantage of the retailers and the e-commerce platform through double discount pricing diminishes, and the profit of the e-commerce platform and the two retailers decreases with the increase in the consumer rebate rate and rebate redemption rate.

Based on the results of the above analysis, we found that the importance of the e-commerce platform rebate rate and the rebate redemption rate for the double discount sales strategy. Our results suggest that e-commerce platforms and retailers should set reasonable rebate rates in conjunction with the actual situation of goods during important discount sales seasons (e.g., internet shopping festivals). Excessive redemption rates can lead to lower profits, in which case if the complexity of consumer access to platform rebates is appropriately set to achieve Pareto improvements in the market.

## 6. Summary

How to effectively determine discount sales strategies and maximize profits for retailers and e-commerce platforms during discount sales and normal sales periods is the key to the sustainable development of e-commerce platforms and retailers during discount sales seasons such as internet shopping festivals. This paper consider the dynamic pricing competition between two vertically differentiated retailers in the retailer discount pricing strategy and the double discount pricing strategy when customers are strategic. We establishes the game models of two pricing strategies and explore the market equilibrium of two retailers and the e-commerce platform, and analyze the effects of the product quality difference coefficient, e-commerce platform rebate rate, consumer rebate redemption rate and level of strategic behavior on retailers' equilibrium pricing and revenue. Our results yield the following insights. Our research show that the optimal pricing of the two periods decreases as the product quality difference increases, and the optimal pricing of the two periods under the double discount pricing strategy is higher than that under the retailer discount pricing strategy, which reveals the retailer's "increase before decrease" routine in the discount sales season. The pricing of the two retailers will decrease with the rebate rate, increase with the rebate cash rate, and increase with the rise in the strategic behavior level in discount sales period. Under the double discount pricing strategy, the demand of high-quality retailers is greater only when the degree of product difference is large, and the demand of low-quality retailers is greater only when the degree of product difference is small. Compared with the retailer discount pricing strategy, double discount pricing strategy can be used to effectively obtain increased profits for both the platform and the retailers and to achieve Pareto improvement of the market. The double discount pricing strategy is often reasonable when consumers have strategic waiting behavior, but the degree of product quality differences, rebate rates, rebate redemption rates and the level of strategic behavior will affect the expected profits of the e-commerce platform and retailers. One of the consequences of excessive rebate activities on the platform is the price increase of retailers in discount sales period, but retailers are pursuing price advantages or at least avoiding price disadvantages. The contradiction between price increases and price advantages has almost entered an endless loop. Overly complex rebate rules for the discount sales season can sap consumers' enthusiasm and affect their interest in shopping. Therefore, retailers should maintain a reasonable degree of product quality differences, and at the same time, e-commerce platforms should optimize their rebate strategies to develop a reasonable rebate ratio according to their own circumstances to achieve a win–win situation with retailers.

In view of the current situation of research in this field and the limitations of this paper, the following areas can be further explored in the future: Consumers have heterogeneous preferences, some consumers may only prefer to purchase from high- or low-quality retailers for various reasons such as preferences for product quality or for channel, online or offline, direct sales or distribution. In the future, the impact of the supply chain power structure on the decision-making of e-commerce platforms and retailers can also be considered. Furthermore, we believe that relaxing some of our assumptions, such as not just two retailers, asymmetric

discount information, etc., has general significance and can deepen our insights, thus therefore constitutes interesting directions for future research.

## Supporting information

**S1 Fig. Diagram of the double discount pricing decision for e-commerce platforms and retailers.**
(TIF)

**S2 Fig. Equilibrium pricing analysis of retailer H under two pricing strategies.**
(TIF)

**S3 Fig. Equilibrium pricing analysis of retailer L under two pricing strategies.**
(TIF)

**S4 Fig. Retailer H demand analysis under two pricing strategies.**
(TIF)

**S5 Fig. Retailer L demand analysis under two pricing strategies.**
(TIF)

**S6 Fig. Comparative analysis of the demand of two retailers under two pricing strategies.**
(TIF)

**S7 Fig. Retailer H profit analysis under two pricing strategies.**
(TIF)

**S8 Fig. Retailer L profit analysis under two pricing strategies.**
(TIF)

**S9 Fig. Comparative analysis of the profit of two retailers under two pricing strategies.**
(TIF)

**S10 Fig. Comparative analysis of the profit of e-commerce platforms under two pricing strategies.**
(TIF)

**S11 Fig. Profit margins between retailers and e-commerce platforms under two pricing strategies.**
(TIF)

**S1 Appendix.**
(DOCX)

## Acknowledgments

Thank you to my supervisor for their help in writing this article, and to my family and friends for their continuous support.

## Author Contributions

**Writing – original draft:** ZHe Chen.

**Writing – review & editing:** Hao Li.

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
