## [Decision Letter · Decision Letter 0]

31 Jul 2023

PONE-D-23-21142How Do E-commerce Platforms and Retailers Implement Discount Pricing Policies under Consumers are Strategic?PLOS ONE

Dear Dr. chen,

Thank you for submitting your manuscript to PLOS ONE. After careful consideration, we feel that it has merit but does not fully meet PLOS ONE’s publication criteria as it currently stands. Therefore, we invite you to submit a revised version of the manuscript that addresses the points raised during the review process.

ACADEMIC EDITOR: *Please provide a final paper with all revisions made and I recommend an additional check on plagiarism and/or compliance with the Journal's guidelines.*

We look forward to receiving your revised manuscript.

Kind regards,

Vincenzo Basile, PhD

Academic Editor

PLOS ONE

“National Social Science Foundation Project (Research on Multi-Party Linkage Mechanism of Car-hailing Market Governance under the Background of Co-construction, Co-Governance and Sharing, 19XGL016, Li Hao).”

3. Please respond by return e-mail with an updated version of your manuscript to include your abstract after the title page.

Reviewers' comments:

Reviewer's Responses to Questions

**Comments to the Author**

1. Is the manuscript technically sound, and do the data support the conclusions?

Reviewer #1: Yes

Reviewer #2: Partly

Reviewer #3: Yes

2. Has the statistical analysis been performed appropriately and rigorously? 

Reviewer #1: Yes

Reviewer #2: No

Reviewer #3: Yes

3. Have the authors made all data underlying the findings in their manuscript fully available?

Reviewer #1: Yes

Reviewer #2: Yes

Reviewer #3: Yes

4. Is the manuscript presented in an intelligible fashion and written in standard English?

Reviewer #1: Yes

Reviewer #2: No

Reviewer #3: Yes

5. Review Comments to the Author

Reviewer #1: This paper constructs a dynamic game model for an e-commerce platform and two retailers when they implement discount pricing, investigate the market equilibrium of the two retailers and e-commerce platform under multiple situations considering consumers’ strategic waiting behavior and competition between the two retailers, and further discuss the effectiveness of retailer discount pricing and the double discount pricing of the platform and retailers. The main results show that the optimal pricing decreases as the difference in product quality narrows under both pricing strategies.

This paper has contributed to the research field of e-commerce supply chain. However, this paper has serious drawbacks, stated as following. Detailed comments are provided below:

The model description of retailer discount pricing strategy (strategy E) is not very clear. I do not understand the retailer discount pricing. Can you demonstrate which parameter denote the discount pricing? Parameter β is the difference quality between the high and low products. So, what’s difference between the period 1 and period 2?

What’s meaning of ν_T in Eq. (1)?

In page 20, why do you assume the parameter settings, i.e., γ=0.4, f=0.7,λ=0.4,c_1=0.05,c_2=0.03,β=0.5,δ=0.2? If you change the values of them, whether the main results will be changed?

The references should follow the format of the Journal.

For S2 Fig (b) and (c), why the p_2^H is near 0? Please give some reasons.

The English level should be improved.

Reviewer #2: In the era of e-commerce evolution, devising sensible discount strategies for e-commerce platforms and retailers is crucial for maintaining a sustainable business. Drawing on the literature related to consumer strategic behavior and discount pricing strategies, this paper constructs a dynamic game model for the e-commerce platform and two retailers. The model investigates market equilibrium under various scenarios, factoring in consumers’ strategic waiting behavior and the impact of retailer discount pricing and dual discounts from the platform and retailers. This is an urgent challenge faced by retailers and e-commerce platforms. Although the author has made substantial efforts, from a professional standpoint, I believe this paper does not yet meet the publication standards of the PLOS ONE journal. Therefore, I recommend a "Reject" decision and provide specific review comments as follows:

1. The paper lacks sufficient innovation and some of the theoretical contributions put forth by the authors do not hold up. To elaborate:

(1) Mu et al. (2020) already investigated the optimal pricing strategy for merchants and e-commerce platforms, taking into account consumer rebates and the dual discount phenomenon. Liu and Zhang (2013) analyzed a market with two firms, H and L, both offering quality-differentiated products with q_H and q_L respectively.

(2) Furthermore, in your discussion of Strategy E, it's unclear how the retailer's offer of a discount is factored in. What is the precise relationship between p_1^i and p_2^i? Is it correct to infer that your paper strictly contemplates whether or not the platform provides rebates?

[1] Mu L, Wang F, Chen L. Research on pricing strategy of E-commerce platform based on strategic consumers[J]. Operations Research and Management Science, 2020, 29(10): 225.

[2] Liu Q, Zhang D. Dynamic pricing competition with strategic customers under vertical product differentiation[J]. Management Science, 2013, 59(1): 84-101.

2. There seem to be issues with your model, with certain expressions appearing ambiguous and confusing.

(1) The range of some parameters needs additional scrutiny. Specifically, "(q_H, q_L) to (1, β) with β∈[0,1]", where β cannot reach the value of 1 as q_H>q_L, and cannot be 0 either, since β might serve as the denominator.

(2) In Strategy E, could you provide further proof to validate that v_2^L fulfills the condition v_2^L=(p_2^L)/β?

(3) The statement "defining the rebate rate of the e-commerce platform as f, consumers can enjoy a discount of (1-f)" is perplexing. The definition of the rebate rate and discount seem rather similar. For instance, if a product is priced at ￥100 and has a rebate rate of "10%", it implies you pay ￥100 and receive ￥10 back, which means you effectively enjoy a discount of 10% and only need to pay ￥90. Consequently, consumers can enjoy a discount of "f", resulting in the actual price paid, (1-f)(p_2^i ) ~, under both platform and retailer discounts. Moreover, typically a discount is deducted directly at the time of purchase, while a rebate is returned to the consumer after purchase at a certain rate, a characteristic that this paper's model fails to represent effectively.

3. The use and citation of references and supporting content in the manuscript is incorrect. The citation marked as [27] in the manuscript states, "We can interpret γ as the level of strategic behavior (Prasad et al. [27]), and a higher γ means……". However, the actual reference is not [27] but [13], which states "Hence, γ can be interpreted as the level of customer’s strategicity/rationality; a higher γ implies that customers are more strategic" (Liu and Zhang 2013). In the manuscript, the authors mention that gamma represents the valuation discount factor for strategic waiting by the parameter, and thus, gamma should only affect value 'v', not '(v-p)'? In addition, strategic customer behavior (i.e., γ) does not play any role in the last period game because there are no future purchase opportunities (Liu and Zhang 2013).

4. The parameter values in the numerical examples lack practical case support, and there should be additional clarification regarding the data sources. The statement that "the optimal pricing of retailer L in the first period decreases before it increases" does not align with Fig. 3(a). The conclusions drawn from the research are quite intuitive, lacking in intriguing findings and valuable managerial insights.

5. Several issues related to English phrasing, grammar, and input errors require a comprehensive review and rectification throughout the text to prevent inaccuracies and increase readability. Therefore, it is recommended that the article undergo a thorough proofreading process. Illustrations of the points that need attention are as follows:

(1) The precision of the language needs to be enhanced. In the "Abstract" section, there are grammatical errors, for example, " In this paper, we constructs a dynamic game……，investigate the……，and further discuss the……" Also, in the "Numerical Study and Managerial Insights" section, the sentence, "The second factor is (the) rebate (rate), which is summarized by the parameter f," needs correction.

(2) Furthermore, some English expressions are challenging to understand. The phrase "Apply for refunding the deposit for presale product after receiving the goods" may not be a suitable example to illustrate the concept of rebate redemption in today's context, as the deposit is typically part of the final payment in reality. Sentences such as "An increase in the rebate rate means that consumers receive fewer discounts from the platform and the retailer will choose to reduce prices to attract consumers " and " The greater the rebate rate is, the smaller the discount on the platform." require refinement.

Reviewer #3: This paper constructs a dynamic game model to investigate the market equilibrium. The research problem is interesting, the model is solid, and the results seems right. I appreciate the authors’ hard work. I propose the following issues, which need the authors’ concerns.

1. The mathematical symbols are ugly. The authors should modify them.

2. I would like to ask why the authors submit this paper to Plos One? It seems that the paper is more suitable to a business journal.

6. PLOS authors have the option to publish the peer review history of their article (what does this mean?). If published, this will include your full peer review and any attached files.

Reviewer #1: No

Reviewer #2: No

Reviewer #3: No

---

## [Author Response · Author response to Decision Letter 0]

25 Oct 2023

For the reviewer's comments, we have attached a document description.

---

## [Editor Report · Decision Letter 1]

26 Oct 2023

PONE-D-23-21142R1How Do E-commerce Platforms and Retailers Implement Discount Pricing Policies under Consumers are Strategic?PLOS ONE

Dear Dr. zhe chen,     Thank you for submitting your manuscript to PLOS ONE. After careful consideration, we feel that it has merit but does not fully meet PLOS ONE’s publication criteria as it currently stands. Therefore, we invite you to submit a revised version of the manuscript that addresses the points raised during the review process.

*Please provide a final paper with all revisions made and I recommend an additional check on plagiarism and/or compliance with the Journal's guidelines.*

We look forward to receiving your revised manuscript.

Kind regards,

Vincenzo Basile, PhD

Academic Editor

PLOS ONE
---

## [Author Response · Author response to Decision Letter 1]

31 Oct 2023

Original Manuscript ID: PONE-D-23-21142

Original Article Title: How Do E-commerce Platforms and Retailers Implement Discount Pricing Policies under Consumers are Strategic?

Dear editor,

Thanks to the editors and reviewers for their comments, which are very helpful in improving the quality of the manuscript. We carefully revised our manuscript, further clarified the logic of writing for improving the quality of the manuscript. The red words on yellow are changes I have made in the original manuscript. Now I response the reviewer's comments with a point by point and highlight the changes in the revised manuscript. Full details of these files are listed below. We sincerely hope that you find our responses and modifications satisfactory and that the manuscript is now acceptable for publication. We are uploading (a) our point-by-point response to the comments (below) (response to reviewers), (b) an updated manuscript(Revised Manuscript with Track Changes), and (c) a clean updated manuscript without highlights.

Thank you and best regards,

Yours sincerely,

Corresponding author：Zhe Chen

E-mail：chenzhe029@163.com

The main corrections in the paper and the responds to the reviewers’ comments are as following:

Reviewer #1:

Comment 1：“The model description of retailer discount pricing strategy (strategy E) is not very clear. I do not understand the retailer discount pricing. Can you demonstrate which parameter denote the discount pricing? Parameter β is the difference quality between the high and low products. So, what’s difference between the period 1 and period 2?”

Response 1：The retail discount pricing strategy in our paper refers to: after a period of sales, retailers will reduce the price of their products, and there will be a price reduction time node. Before the node, we define it as Period 1, which is the normal sales period. At this time, the normal sales price is , after the node, we define it as Period 2, which is the price reduction sales period, where the discount price is . We have demonstrated through numerical simulation that the sales price of Period 1 is higher than that of Period 2, that is, , which means that consumers can enjoy discounts . For example, the Chinese shopping festival "Double Eleven" refers to a price reduction sale starting on November 11th. Before November 11th, the price of the Arthur ASICS brand sports shoe Flux 4 was 559 RMB. When the product starts to be reduced to 332 RMB at 00:00 on November 11th, consumers can enjoy a 41% discount. In the last paragraph of “The Model”, we analyzed in detail the meaning of discount pricing.

Comment 2：“What’s meaning of ν_T in Eq. (1)?”

Response 2：In equation 1, refers to the undifferentiated point estimation of the utility of a product purchased by a consumer in two periods, where the utility of a product purchased by a consumer in a normal sales period and a discounted sales period is equal, that is, , simplified to: . We have explained this in detail in the second paragraph after “Proposition 1”.

Fig.1

Comment 3： “In page 20, why do you assume the parameter settings, i.e., γ=0.4, f=0.7,λ=0.4,c_1=0.05,c_2=0.03,β=0.5,δ=0.2? If you change the values of them, whether the main results will be changed? ”

Response 3：Regarding parameter assumption: We assume that , , , , ， is randomly selected to examine the relationship between factors. In fact, the size of the parameter does not change the equilibrium result. We change the value of to , and keep the other parameter values fixed , , , ， . This will correspondingly change the specific values such as the main price, but the main trend remains unchanged. Similarly, we only change the value of one parameter and keep the other values unchanged to find the same trend. In addition, we also analyzed the impact of other random parameters on the equilibrium results in the article. In order to better present the results, only one set of values was selected in this article. Fig. 2 is the original graph, and Fig. 3 is the graph after parameter changes. It can be seen that although the values have changed, the overall trend has not changed. Based on this, we have come to a more universal conclusion. We point out this point in the first paragraph of “Numerical Study and Managerial Insights”.

Fig. 2 Fig.3

Comment 4：“The references should follow the format of the Journal.”

Response 4：Thank you very much for pointing out this issue. We are very sorry about the reference format and strictly follow the relevant format for modification. We have made every effort to meet the required editorial correction standards.

Comment 5： “For S2 Fig (b) and (c), why the p_2^H is near 0? Please give some reasons.”

Response 5：Regarding this issue, this is because the research conclusion of this article is similar to the Bertrand Nash game, which means that due to price competition between oligarchs, the final result is the same as in a completely free competition market, that is, the price drops to equal the cost. The price approaching 0 in the figure is obtained through numerical examples. The reason for this result is that under discounted pricing, the price competition between duopolies is carried out to attract more consumers. In order to attract more consumers, retailers engage in oligopolistic price competition without providing discounts on the platform. Additionally, due to the assumption of very low costs, the price in the second phase of the numerical simulation tends to be very low and seems to be zero. We point out this point in the second paragraph of the “Optimal Pricing Strategy Analysis” in the article.

Comment 6：“The English level should be improved. ”

Response 6：We are very sorry for the issue with our English proficiency. We have read through the entire text and have further revised any areas we consider unreasonable. Thank you for your sincere suggestions. 

Dear reviewers, 

Thank you for your careful review and constructive suggestions regarding our manuscript. We have revised the manuscript in accordance with the comments and marked all the amends on our revised manuscript. We hope that the modifications can be approved. Thank you again for your comment. We are happy to answer any further questions and comments you may have.

Thank you again and best regards.

Sincerely,

Hao Li, Zhe Chen

Reviewer #2:

Comment 1：1.“The paper lacks sufficient innovation and some of the theoretical contributions put forth by the authors do not hold up. To elaborate:

(1) Mu et al. (2020) already investigated the optimal pricing strategy for merchants and e-commerce platforms, taking into account consumer rebates and the dual discount phenomenon. Liu and Zhang (2013) analyzed a market with two firms, H and L, both offering quality-differentiated products with q_H and q_L respectively.

(2) Furthermore, in your discussion of Strategy E, it's unclear how the retailer's offer of a discount is factored in. What is the precise relationship between p_1^i and p_2^i? Is it correct to infer that your paper strictly contemplates whether or not the platform provides rebates?

[1] Mu L, Wang F, Chen L. Research on pricing strategy of E-commerce platform based on strategic consumers[J]. Operations Research and Management Science, 2020, 29(10): 225.

[2] Liu Q, Zhang D. Dynamic pricing competition with strategic customers under vertical product differentiation[J]. Management Science, 2013, 59(1): 84-101.”

Response 1：(1) We are very sorry that we may not have made a clear comparative analysis with existing literature. We pointed out this point in the “Literature Review”. In fact, our article is very different from the two papers you mentioned. Mu et al. (2020) did study the optimal pricing strategies of merchants and e-commerce platforms considering consumer rebates and double discounts. However, this article assumes that a market system consisting of an online retailer and an e-commerce platform does not consider the competitive relationship between high and low quality retailers. Moreover, this article studies whether e-commerce platforms provide rebates. This study is more in line with the reality of competition in the platform, and delves into the comparison of two strategies of discount pricing for retailers and dual discount pricing for retailers and e-commerce platforms, further expanded the research of Mu et al.

Liu and Zhang (2013) studied the dynamic pricing strategies of two companies, H and L, that provide differentiated quality qH and qL products. They compared static and dynamic pricing strategies between retailers, but did not consider the impact of e-commerce platforms on retailers; This article not only considers two retailers with vertical differentiation in the same competitive platform, but also considers the different discount pricing strategies of retailers and e-commerce platforms. The competition between retailers and platform rebates have a significant impact on the optimal strategy selection.

(2) We are very sorry that due to our unclear statement, the retailer discount we are considering is displayed through a two cycle price reduction. After normal sales, retailers will lower the product price. We define the time point before the price reduction as cycle 1, which is the normal sales cycle. At this time, the sales price is ; We define the price reduction node as cycle 2, which is the price reduction sales cycle. At this time, the discount price is . We have demonstrated through numerical simulation that the sales price for period 1 is higher than that for period 2, ,which is a discount pricing strategy with a decrease in price. As shown in the figure below, it can be seen that （ ） is true. We pointed out this in the last paragraph of “The Model”.

This article considers the rebate strategy of e-commerce platforms, which make decisions by providing the profit margin before and after the rebate. For example, platforms such as JD.com and Taobao do not offer rebate discounts on a daily basis, but offer certain rebates during holidays or shopping festivals. From a practical perspective, platforms offer rebates or different types of rebates. Therefore, this article considers the platform's decision on whether to offer rebates based on actual situations.

Fig.1

Comment 2：2. There seem to be issues with your model, with certain expressions appearing ambiguous and confusing.

(1) The range of some parameters needs additional scrutiny. Specifically, "(q_H, q_L) to (1, β) with β∈[0,1]", where β cannot reach the value of 1 as q_H>q_L, and cannot be 0 either, since β might serve as the denominator.

(2) In Strategy E, could you provide further proof to validate that v_2^L fulfills the condition v_2^L=(p_2^L)/β?

(3) The statement "defining the rebate rate of the e-commerce platform as f, consumers can enjoy a discount of (1-f)" is perplexing. The definition of the rebate rate and discount seem rather similar. For instance, if a product is priced at ￥100 and has a rebate rate of "10%", it implies you pay ￥100 and receive ￥10 back, which means you effectively enjoy a discount of 10% and only need to pay ￥90. Consequently, consumers can enjoy a discount of "f", resulting in the actual price paid, (1-f)(p_2^i ) ~, under both platform and retailer discounts. Moreover, typically a discount is deducted directly at the time of purchase, while a rebate is returned to the consumer after purchase at a certain rate, a characteristic that this paper's model fails to represent effectively.

Response 2：(1) Thank you very much for pointing out this issue. We have indeed overlooked this issue before, and your suggestion has made our model more rigorous. We have also made modifications to it, and we explained this in “The Model”.

(2) Let , at this point, the consumer valuation that retailer L can sell as far as possible in Period 2 is satisfied , therefore is determined by the value of and . At this time, only retailer L's products have demand in the market.

(3) We are very sorry that our statement may have caused ambiguity. In China, offering a few discounts is equivalent to enjoying a few discounts. In fact, what you said is quite correct. It should be that when the actual payment price is , consumers enjoy a discount . Secondly, after comprehensive consideration, we have decided to directly reflect the rebates paid by e-commerce platforms on consumer payment prices to more intuitively demonstrate the changes in consumer spending. We explained this in “Analysis of the double discount pricing strategy for retailers and e-commerce platforms”.

Comment3：3. The use and citation of references and supporting content in the manuscript is incorrect. The citation marked as [27] in the manuscript states, "We can interpret γ as the level of strategic behavior (Prasad et al. [27]), and a higher γ means……". However, the actual reference is not [27] but [13], which states "Hence, γ can be interpreted as the level of customer’s strategicity/rationality; a higher γ implies that customers are more strategic" (Liu and Zhang 2013). In the manuscript, the authors mention that gamma represents the valuation discount factor for strategic waiting by the parameter, and thus, gamma should only affect value 'v', not '(v-p)'? In addition, strategic customer behavior (i.e., γ) does not play any role in the last period game because there are no future purchase opportunities (Liu and Zhang 2013).

Response 3：Thank you very much for your correction of the literature citation. The essence of it should be as you said, and we have made modifications to it. But unlike Liu, what we define as the utility discount factor, like Levin Y, McGill J, and Nediak M, represents the degree of consumer strategy. A larger value indicates that consumers have greater utility in cross period purchases, and a higher willingness to wait until period 2 (discounted sales period) to make purchases. We are studying the relationship between period 1 and period 2. And we explained this in “The Model”.

[1] Levin Y , Mcgill J , Nediak M .Optimal dynamic pricing of perishable items by a monopolist facing strategic consumers[J].Operations Research, 2010.

[2] Levin Y , Mcgill J , Nediak M .Dynamic Pricing in the Presence of Strategic Consumers and Oligopolistic Competition[J]. 2017.DOI:10.1287/mnsc.1080.0936.

Comment4：4. The parameter values in the numerical examples lack practical case support, and there should be additional clarification regarding the data sources. The statement that "the optimal pricing of retailer L in the first period decreases before it increases" does not align with Fig. 3(a). The conclusions drawn from the research are quite intuitive, lacking in intriguing findings and valuable managerial insights.

Response 4：Regarding parameter assumption: We assume that , , , ， is randomly selected to examine the relationship between factors. In fact, the size of the parameter does not change the equilibrium result.

We change the value to , we keep the other parameter values fixed , 。 ， , This will correspondingly change the specific values such as the main price, but the main trend remains unchanged. Similarly, we only change the value of any other parameter and keep the other values unchanged. In multiple experiments, we found that changing the numerical simulation values will not affect the trend direction. Therefore, we chose a set of data that can most intuitively display the changing trend. Fig. 2 is the original graph, and Fig. 3 is the graph after parameter changes. Although the parameter values have changed, the overall trend in Fig. 2 and 3 has not changed. We point out this point in the first paragraph of “Numerical Study and Managerial Insights”.

 Fig.2 Fig.3

Thank you for your correction. Due to our mistake, we did not accurately describe the price changes of retailers in Cycle 1. We have now checked and corrected several issues related to English wording, grammar, and input errors throughout the text, and have proposed some more thought-provoking discoveries and valuable management insights.

Comment 5：5. Several issues related to English phrasing, grammar, and input errors require a comprehensive review and rectification throughout the text to prevent inaccuracies and increase readability. Therefore, it is recommended that the article undergo a thorough proofreading process. Illustrations of the points that need attention are as follows:

(1) The precision of the language needs to be enhanced. In the "Abstract" section, there are grammatical errors, for example, " In this paper, we constructs a dynamic game……，investigate the……，and further discuss the……" Also, in the "Numerical Study and Managerial Insights" section, the sentence, "The second factor is (the) rebate (rate), which is summarized by the parameter f," needs correction.

(2) Furthermore, some English expressions are challenging to understand. The phrase "Apply for refunding the deposit for presale product after receiving the goods" may not be a suitable example to illustrate the concept of rebate redemption in today's context, as the deposit is typically part of the final payment in reality. Sentences such as "An increase in the rebate rate means that consumers receive fewer discounts from the platform and the retailer will choose to reduce prices to attract consumers " and " The greater the rebate rate is, the smaller the discount on the platform." require refinement.

Response 5：Thank you for your suggestion. We have thoroughly proofread the article. Corrected grammar and sentence errors, and rewrote sentences that were not clearly expressed, refining them based on actual situations. The reason why we say "Apply for refunding the deposit for presale product after receiving the goods" is because the promotion methods on the platform are more diverse nowadays. For example, during the Tmall Double 11 shopping festival, we launched a deposit first reservation product. After the final payment is settled, consumers can directly apply for a refund deposit on the shopping details page to complete the rebate.

Dear reviewer, 

Thank you very much for taking the time to review this manuscript. We really appreciate your feedback and suggestions. We have revised the manuscript based on your comments and marked all modifications on the revised manuscript. We look forward to receiving your letter. If you have any questions, please do not hesitate to contact me. 

Thank you again and best regards.

Sincerely,

Hao Li, Zhe Chen

Reviewer #3: 

Comment 1： 1. The mathematical symbols are ugly. The authors should modify them.

Response 2：Thanks for your interest in our research. We tried our best to modified study for making article meet the requirement. We have optimized the entire article and made adjustments according to the requirements of “PLOS ONE” journal.

Comment 2：2. I would like to ask why the authors submit this paper to Plos One? It seems that the paper is more suitable to a business journal.

Response 2：Thank you for your sincere suggestion. PLOS ONE accepts articles in the fields of humanities and social sciences. Our article elaborates on discount sales seasons such as the Internet Shopping Festival, which essentially belongs to the field of humanities and social sciences. This article involves solving complex models with multiple cycles in derivation, exploring the economic laws under price discounts, revealing the social phenomenon of "increasing prices first and then decreasing prices", and providing certain guidance for the pricing of e-commerce platform retailers and consumer purchasing decisions. Based on this consideration, we believe that this article is suitable for submission to PLOS ONE.

Dear reviewer, 

We are grateful for your effort reviewing our paper and your positive feedback. We have read through the entire text and revised the manuscript according to the journal's requirements based on your feedback, and have marked all modifications on the revised manuscript.

Thank you and best regards.

Sincerely,

Hao Li, Zhe Chen

---

## [Editor Report · Decision Letter 2]

18 Dec 2023

How Do E-commerce Platforms and Retailers Implement Discount Pricing Policies under Consumers are Strategic?

PONE-D-23-21142R2

Dear Dr. zhe chen,

We’re pleased to inform you that your manuscript has been judged scientifically suitable for publication and will be formally accepted for publication once it meets all outstanding technical requirements.

Kind regards,

Vincenzo Basile, PhD

Academic Editor

PLOS ONE

Additional Editor Comments:

*Please provide a final paper with all revisions made and I recommend an additional check on plagiarism and/or compliance with the Journal's guidelines.*

---

## [Editor Report · Acceptance letter]

1 Mar 2024

PONE-D-23-21142R2 

PLOS ONE

Dear Dr. Chen, 

I'm pleased to inform you that your manuscript has been deemed suitable for publication in PLOS ONE. Congratulations! Your manuscript is now being handed over to our production team.

Kind regards, 

on behalf of

Dr. Vincenzo Basile 

Academic Editor

PLOS ONE